

# Phylogenetically-controlled correlates of primate blinking behaviour

Sean A. Rands

School of Biological Sciences, University of Bristol, Bristol, UK

## ABSTRACT

Eye blinking is an essential maintenance behaviour for many terrestrial animals, but is also a risky behaviour as the animal is unable to scan the environment and detect hazards while its eyes are temporarily closed. It is therefore likely that the length of time that the eyes are closed and the length of the gap between blinks for a species may reflect aspects of the ecology of that species, such as its social or physical environment. An earlier published study conducted a comparative study linking blinking behaviour and ecology, and detailed a dataset describing the blinking behaviour of a large number of primate species that was collected from captive animals, but the analysis presented did not control for the nonindependence of the data due to common evolutionary history. In the present study, the dataset is reanalysed using phylogenetic comparative methods, after reconsideration of the parameters describing the physical and social environments of the species. I find that blink rate is best described by the locomotion mode of a species, where species moving through arboreal environments blink least, ground-living species blink most, and species that use both environments show intermediate rates. The duration of a blink was also related to locomotion mode, and positively correlated with both mean species group size and mean species body mass, although the increase in relation to group size is small. How a species moves through the environment therefore appears to be important for determining blinking behaviour, and suggests that complex arboreal environments may require less interruption to visual attention. Given that the data were collected with captive individuals, caution is recommended for interpreting the correlations found.

## INTRODUCTION

Eye blinking, where both eyes are temporarily closed by movements of the eyelids, is a behaviour that is performed continuously and frequently by humans and other animals (*Blount, 1927*; *Walls, 1942*). As well as being an essential maintenance behaviour that ensures that the cornea surface is lubricated and clean (*Nakamori et al., 1997*), blinking behaviour may also be linked to how an individual processes and responds to both environmental and endogenous stimuli. Human studies (*Drew, 1951*; *Holland & Tarlow, 1972*; *Goldstein, Bauer & Stern, 1992*; *Leal & Vrij, 2008*, *2010*) have demonstrated that blinking decreases when cognitive demand increases, and blinking behaviour may be tied

Corresponding author
Sean A. Rands,
sean.rands@bristol.ac.uk

with non-verbal communication (*Cummins, 2012*), potentially differing according to the identity of the individual that communication is directed towards (*Descroix et al., 2018*). The timing and frequency of blinks may also be controlled to avoid missing important visual information (*Nakano et al., 2009*; *Shin et al., 2015*; *Wiseman & Nakano, 2016*; *Maffei & Angrilli, 2019*; *Ranti et al., 2020*). Spontaneous eye blinks cause the individual to momentarily lose visual information, but in humans this 'blackout' (where the eyes are closed and it is therefore not possible to collect visual information) is not perceived by the individual due to an attentional suppression mechanism (*Volkmann, Riggs & Moore, 1980*; *Riggs, Volkmann & Moore, 1981*), with a suppression of activity in both the visual cortex and other areas of the brain that are involved with awareness of changes in the environment (*Bristow et al., 2005*).

Fewer studies have explored how blinking is related to other behaviour in non-human species. Blinking may be tied with attentional state, as is demonstrated by the reduction in blinking that occurs during sleep in herring gulls *Larus argentatus* (*Amlaner & McFarland, 1981*), which, like other birds and aquatic mammals (*Kendall-Bar et al., 2019*; *Rattenborg, Lima & Amlaner, 1999*), can exhibit sleep states where one eye is kept open. Blinking behaviour may also be mediated by risky or stressful situations, which have been shown to cause blink rates to decrease in both American crows *Corvus brachyrhynchos* (*Cross et al., 2013*), horses *Equus caballus* (*Merkies et al., 2019*), and peafowl *Pavo cristatus* (*Yorzinski, 2016*). For species that experience predation, blinking may be a risky behaviour as it momentarily stops the individual from collecting visual information about the presence of predators in its environment. If an animal is social, it can partly rely on other individuals to be vigilant for predators, which would mean that a group-living animal could afford to blink more often than one that has to monitor the environment on its own. This reduction in blinking when in small groups or alone has been suggested for both olive baboons *Papio anubis* (*Matsumoto-Oda et al., 2018*) and chickens *Gallus gallus* (*Beauchamp, 2017*), and the individual's rate of blinking has been shown to be positively correlated with group size in red deer *Cervus elaphus* (*Rowe, Robins & Rands, 2020*). Blink behaviour may be tied with the behaviour the animal is conducting, as demonstrated by the reduction in blinking during flight behaviour by grackles *Quiscalus mexicanus*, which may occur in order to reduce the risks of a dangerous collision with an unobserved object during flight (*Yorzinski, 2020b*). Evidence also suggests that blinks are timed to occur alongside head movement behaviours in both rhesus macaques *Macaca mulatta* (*Gandhi, 2012*) and peafowl (*Yorzinski, 2016*), where the animal is simultaneously moving its head during a blinking episode (and the quality of information received from the environment is assumed to be reduced if the eyes were open during this movement). These diverse animal examples demonstrate that blinking behaviour is potentially influenced by both the environmental and the social conditions that an animal experiences.

Given that the blinking behaviour of a species will be related to its ecology and life history, we would expect there to potentially be wide variation in this behaviour between species. The handful of comparative studies that have been conducted demonstrate that there is variation in blinking behaviour in both mammals (*Stevens & Livermore, 1978*;

*Tada et al., 2013*) and birds (*Kirsten & Kirsten, 1983*). *Tada et al. (2013)* presented a comprehensive dataset that collected the blink rate and interblink intervals for 71 named species and subspecies of primate, collected using a standardised technique. The study correlated these rates and intervals with ecological characteristics of the species, and presented results that suggested that diurnal species blinked more than nocturnal ones, that habitat type (whether a species was arboreal), semi-arboreal or ground-living (denoted 'terrestrial' in the original paper) was not linked to blinking, that eye-blink rate and interblink interval were positively correlated with mean species body mass, and that blink rate was positively correlated with group size.

These results could potentially give us some clues about the links between blinking and an animal's ecology and life history. However, the analyses presented are problematic as they did not control for phylogenetic similarity. Species that are closely related are likely to show similar behaviours and traits due to this recent history, and therefore any analyses that compare species need to account for this nonindependence of the datapoints (*Felsenstein, 1985*; *Harvey & Pagel, 1991*). It is therefore possible that the trends reported from the analysis of the blinking dataset may be an artefact of considering data from closely-related species (which can be seen in the panels of figure 4 in the original article, where differently coloured symbols representing different families are often tightly bunched together; this bunching together is likely to be because these closely-related species share similar adaptations because they share a recent common ancestor, which means that the measures recorded for their trait are not as independent of each other as if we were to compare more distantly-related species). However, some excellent estimates of primate phylogeny exist (*Nunn & Barton, 2001*), and so it is possible to reanalyse these data controlling for common ancestry. Here, I present a phylogenetically-controlled reanalysis of the *Tada et al. (2013)* data that uses a well-established model of primate evolution, with the intention of reaffirming whether aspects of a species' ecology are related with a species' blinking behaviour.

*Tada et al. (2013)* collected several different measures of species blinking behaviour. Two of these (blink rate, which counts the number of blinks during a period of time, and blink duration, which measures the mean length of an isolated blink) measure aspects of blinking behaviour that are likely to be reduced in situations where it is important to not miss information from the environment that may change rapidly (e.g. the sudden appearance of a predator or a food item, or the location of a stable location to place a hand or foot when moving through a complex environment). The species sampled occupy a range of different habitats that differ in their complexity, but can be broadly divided into those that spend most of their lives in an arboreal environment, those that spend most of their lives on the ground, and those that divide their time between the two. It is likely that it is more immediately dangerous for a primate to misjudge a movement decision in an arboreal environment, and we can therefore predict that both blink rate and blink duration are going to be reduced in species that spend their lives in trees. *Tada et al. (2013)* did not find any measure of blinking to be correlated with the movement environment (referred to as 'habitat type' in their paper), although the figures presented in
their paper suggest that there may be a trend. Here, I present phylogenetically-controlled analyses that test this prediction.

A third measure (the isolated blink ratio, which measures the proportion of time that a blink occurs independently of any other body movement) was also considered by *Tada et al. (2013)* as a method for assessing whether blinking is synchronous with other behaviours that could reduce the animal's ability to sample information from the environment (as considered in *Gandhi (2012)* and *Yorzinski (2016)*), and would be expected to decrease in situations where not missing information is important. However, the description of how the criteria under which this last measure was collected lacks sufficient detail to be able to fully characterise or replicate the measure in a fresh study, and so I do not consider the 'isolated blink ratio' data within the current reanalysis.

Alongside exploring whether movement environment influences blinking behaviour, I also consider a number of other ecological factors that could be tied with blinking behaviour. The analysis by *Tada et al. (2013)* suggests that group size may influence blinking behaviour, although it is unclear whether living in larger groups should lead to less blinking behaviour (i.e. to better monitor social information) or more (i.e. if blinking were to be involved in social communication). Similarly, species showing diurnal behaviour blinked more than nocturnal species, which suggests that blinking may be reduced when light levels are low. Mean body size is also considered here as *Tada et al. (2013)* found it to be correlated with other variables considered, although it is less clear how body size should influence blink rate. Finally, I also consider whether the trophic requirements of a species influences its blinking behaviour, as differing trophic styles may require differing amounts of attention in order to locate and handle food items within the environment. The results will assess both the importance of controlling for phylogenetic similarity when considering the evolution of behaviour, and whether blinking behaviour could reveal new insights into the ecology of species.

## METHODS

The data presented in Table 1 of *Tada et al. (2013)* were used. This original dataset contains three measures of blinking for 71 named species and subspecies of primate, along with a number of explanatory variables. The three separate measures, each summarising observations for between one and six individuals of a species, consisted of the blink rate (in blinks per minute), blink duration (the mean length of an individual blink, in ms), and the isolated blink ratio, which was discussed above and is not considered further in the current analysis. Each individual recorded provided at least five minutes of video footage where the eyes were visible: full details of the data extraction are given in detail in *Tada et al. (2013)*.

As well as providing these data describing blinking, *Tada et al. (2013)* also provided descriptive measures of the activity, habitat type, mean species group size and mean species body mass, using previously published data that they harvested from the literature. In the current analysis, I use this collated data for mean species group size and mean species body mass (in kg, noting that the data used in *Tada et al. (2013)* correlate strongly with equivalent mean species mass data from *Galán-Acedo et al. (2020)*,

Pearson correlation of logged masses $r = 0.988$, $t_{64} = 50.24$, $p < 0.001$, acknowledging that the latter does not contain values for *Cebus apella* or *Callicebus moloch*, and taking the mean value for cases in *Galán-Acedo et al. (2020)* where more than one mean mass is given for a species). I also used the original data for activity, which records individuals as either being primarily diurnal or nocturnal in behaviour. This categorised five species as being nocturnal, and the other 66 as diurnal; ideally, the two species of *Eulemur* should be scored as cathemeral instead (*Donati & Borgognini-Tarli, 2006*; *Galán-Acedo et al., 2020*), but this would have given a category containing only two datapoints, and so these are still considered here as being within the original 'diurnal' category.

The dataset given by *Tada et al. (2013)* also considered a measure of habitat as a descriptor for blinking. This classification of habitat considered whether a species was primarily arboreal, semi-arboreal or 'terrestrial'. I did not use this measure as it did not fully account for the ecological habitat that a species was normally found in, and scoring habitat is complicated because most of the species considered are not exclusively found in one habitat type (and most of the species can be found in forest habitats), as can be seen for the considered species in the data collected by *Galán-Acedo et al. (2020)*. The original habitat classification considered whether a species is primarily found in trees or on the ground, which could be taken as a classification of its principal locomotion, but it is not clear exactly what criteria were being used for defining this for a species. In order to give a rigorous definition of the locomotion mode of each species, I instead collected data from *Galán-Acedo et al. (2020)* that described whether a species was primarily ground-living, arboreal, or moved between both the ground and the arboreal environments (the full dataset presented in *Galán-Acedo et al. (2020)* is described in detail by *Galán-Acedo et al. (2019)*). This gave a set of descriptors that did not fully correspond to the original habitat description by *Tada et al. (2013)*, with 16 (23.5%) described differently. Trophic guild data were also collected from *Galán-Acedo et al. (2020)*, and classified the species as frugivores ($n = 29$), folivores ($n = 4$), joint foli- and frugivores ($n = 9$), gummivores ($n = 4$), and omnivores ($n = 22$).

The species list presented in *Tada et al. (2013)* was used to create a phylogeny, using *10kTrees* version 3 (*Arnold, Matthews & Nunn, 2010*, available at https://10ktrees.nunn-lab.org/Primates): the phylogeny I used was a chronogram generated with a consensus of 10,000 simulated trees. Taxonomic differences existed between the site and the data presented in *Tada et al. (2013)*, and the following substitutions were made to comply with the online species list: *Varecia variegata variegata* was used instead of *V. variegata*; *Eulemur fulvus fulvus* instead of *E. fulvus*; *E. macaco macaco* instead of *E. macaco*; *Callithrix pygmaea* instead of *Cebuella pygmaea*; *Saguinus mystax* instead of *S. labiatus*; *Semnopithecus entellus* instead of *Presbytis entellus*; and *Pan troglodytes troglodytes* instead of *P. troglodytes*. *Cercopithecus mitis*, *Colobus polykomos angolensis*, and *Cercocebus torquatus lunulatus* were considered at the species level, as subspecies were not differentiated in the database. Where subspecies were substituted for species, the decision made was arbitrary, but was unlikely to affect the phylogeny generated as no conflicting sister subspecies were present in the dataset. *Cercopithecus ascanius schmidti* and *Macaca fuscata yakui* were presented in the original dataset as different datapoints

**Table 1  Minimal adequate model describing blink rate.**

|  | Estimate | Standard error | t | p |
|---|---|---|---|---|
| Intercept | 2.13 | 0.07 | 3.25 | 0.002 |
| Locomotion: both arboreal and ground-living | 0.81 | 0.27 | 3.05 | 0.003 |
| solely ground-living | 0.81 | 0.40 | 2.05 | 0.044 |

Note:
Estimated coefficients given for sqrt(blinkrate+1) ~ locomotion, where 'solely arboreal' locomotion represents the baseline for the term. Adjusted $r^2$ = 0.102, residual SE = 0.466 on 65 df, $F_{2,65}$ = 4.79, $p$ = 0.011.

from *C. ascanius* and *M. fuscata*, but were not differentiated from the species within the tree database, and so I decided to remove the subspecies data from the dataset, keeping only the datapoints for the named species. Similarly, *Cercocebus chrysogaster* was assumed to be a synonym for *C. agilis*, and so the data presented for *C. chrysogaster* was also removed from the dataset. This left a dataset with 68 species, and an associated ultrametric consensus tree.

I fitted phylogenetic generalised least square (phylogenetic GLS) models (*Pagel, 1999*; *Freckleton, Harvey & Pagel, 2002*), assuming a Brownian motion model of trait evolution. This was done using *caper* 1.0.1 (*Orme et al., 2018*) within *R* 4.0.2 (*R Development Core Team, 2020*), after initially confirming that the three measures showed phylogenetic signal using *phytools* 0.7–47 (*Revell, 2012*). In order to improve the fit of the data to the phylogenetic GLS models used, I specified that the *pgls* function optimised branch lengths for each model using standardised transformations of the covariance matrix (*Pagel, 1997, 1999*; *Orme et al., 2018*): for blink rate, κ was optimised by maximum likelihood methods, and δ was optimised for blink duration. Both of the behavioural measures were initially tested using the model "locomotion + mass + group size + trophic guild + diurnality", and the least significant terms were then removed sequentially until the minimal adequate model was found (at each removal, the model simplification process was checked using likelihood ratio tests, and the simpler of the current and previous iteration of the model was retained when the likelihood ratio was non-significant). To satisfy assumptions of residual normality, the blink rate was square root transformed for all models. Full code, along with the dataset used and the consensus tree, is presented in the Supplemental Material.

## RESULTS

Blink rate was best described by locomotion mode alone (Table 1). Blink rate was lowest in arboreal, and highest in those that were solely ground-living or that spent time on both the ground and in the arboreal habitat (which present near-identical estimates of blink rate in Table 1, and so are unlikely to differ) (Fig. 1A).

Blink duration was best described using 'locomotion + mass + group size' (Table 2). Blink duration was similar in solely ground-living and solely arboreal species (Table 2; Fig. 1B), and shorter in those that normally switched between ground-based and arboreal locomotion. Blink duration lengthened with increases in both body mass and group size (Table 2), but the increase with group size (1.03) is very shallow.

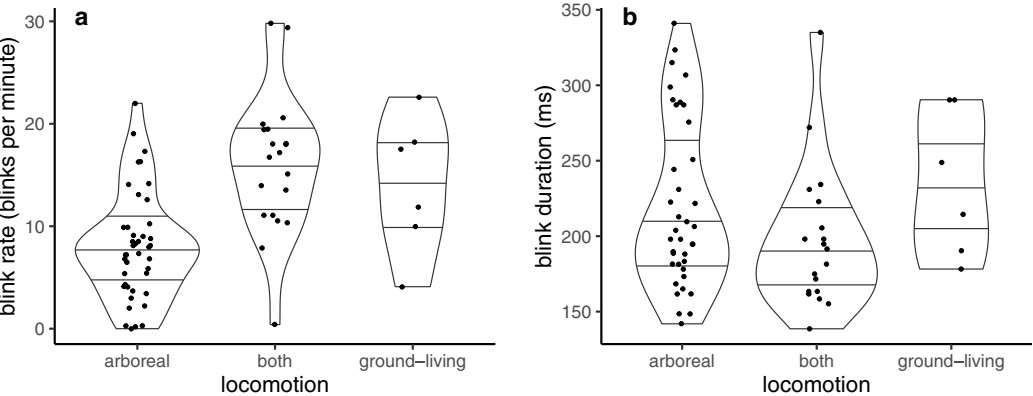

**Figure 1 Violin plots showing probability densities describing how (A) blink rate and (B) blink duration are related to locomotion mode.** Lines within the shapes represent the median and inter-quartile ranges of the data.                                                  

**Table 2 Minimal adequate model describing blink duration.**

|  | Estimate | Standard error | $t$ | $p$ |
|---|---|---|---|---|
| Intercept | 225.20 | 35.26 | 6.37 | <0.001 |
| Locomotion: both arboreal and ground-living | −48.24 | 17.21 | −2.80 | 0.007 |
| solely ground-living | 20.65 | 28.04 | 0.74 | 0.464 |
| Mean body mass | 1.40 | 0.60 | 2.34 | 0.022 |
| Mean group size | 1.03 | 0.43 | 2.38 | 0.021 |

Note:
Estimated coefficients given for duration ~ locomotion + body mass + group size, where 'solely arboreal' locomotion represents the baseline for the term. Adjusted $r^2$ = 0.307, residual SE = 0.147 on 58 df, $F_{4,58}$ = 7.88, $p < 0.001$.

## DISCUSSION

The analysis presented here suggests that both blink rate and blink duration are related to locomotion mode in primates. Blink rate is lowest in arboreal species, and higher in species that spend some (or all) of the time moving on the ground, while blink duration is shortest in species that switch between the ground and arboreal habitats, and longer in those that spend their lives either moving solely on the ground or solely in the trees. This suggests that arboreal species may need to pay more attention to the complex environment that they live in by reducing the overall number of blinks that they conduct, but it is species that use both environments that keep blinks short, possibly because of the increased number of hazards that may be encountered by moving in multiple habitats (as is echoed in the intraindividual changes in behaviour shown by grackles (*Yorzinski, 2020b*) when engaged in risky flight behaviours). These results contradict the previous finding (*Tada et al., 2013*) that habitat was not correlated with blinking behaviour (where habitat was represented by a similar but not identical measure to the locomotion data used here). Because human studies (*Drew, 1951*; *Holland & Tarlow, 1972*; *Goldstein, Bauer & Stern, 1992*; *Leal & Vrij, 2008*, *2010*) have demonstrated that blinking is reduced during tasks with a high cognitive demand, it is tempting to infer from the current

results that tree-dwelling species blink less than ground-living species due to the risks involved with moving through a complex three-dimensional arboreal habitat, as suggested above. However, the original data were collected by videoing captive individuals, and it is unclear (and unlikely) whether any of the arboreal species were moving through a suitably complex environment during the data collection. It would be enlightening to quantify blink rates whilst individuals were moving through the environment that they were adapted to. Blink rate was lowest in species that spent at least some of their lives in trees, but blink duration was shortest for the species that mixed their time between arboreal and ground environments. These results are potentially contradictory, as solely arboreal species will conduct fewer blinks, but of longer duration, which could mean that there is little difference between species in the total time spent with the eyes closed. Whether this is important depends upon which aspects of blinking behaviour are important for a species, and I would recommend careful consideration of whether these different aspects of blinking are open to different selection pressures. Another reason why blink rate and blink duration give different results is that there may be differing degrees of accuracy in measuring these from video recordings. Measuring blink rate relies on being able to characterise a single blink episode, whereas measuring the duration of an individual blink requires recording a video that has a high rate of image capture that will allow the observer to accurately identify both when a blink starts and ends and its length. If the video recording has a low frame rate (meaning fewer images are captured per second), or insufficient image quality, then the blink duration data will be extremely coarse, and may not tell us much about an animal's behaviour. This could partly explain why the blink rate and blink duration results differ in the current study, and it is recommended that any experiments attempting to measure blinking consider the limitations that their data recording will impose on the measures of blink duration that they intend to extract from these data.

The data also suggest that blink duration increases very slightly with mean species group size, confirming the original result from *Tada et al. (2013)*. The reanalysis I present here did not show that blink rate was related to group size, unlike the original analysis. However, both an increased rate and increased duration could lead to the eyes being closed for a longer total period of time. Standard theory considers how vigilance levels are influenced by size of group that an individual is associated with, where individual vigilance will decrease as the number of co-vigilant group members increases (*Pulliam, 1973*; *Elgar, 1989*; *Beauchamp, 2010*). This leads to the prediction that the total amount of time spent with eyes closed should increase with group size, echoing the results presented here. However, caution should be attached to the present result, as the estimated increase is very small, and also that these predictions relate to differences in group size within species, rather than comparisons of differing mean group sizes for different species. Tests of how blinking relates to different group sizes should be tested within species, as has been demonstrated in red deer and chickens (*Rowe, Robins & Rands, 2020*; *Beauchamp, 2017*). The relationship seen with group size may also be correlated with species body mass (as was suggested by *Tada et al. (2013)*, and confirmed here for blink duration), and is

likely to be due to the complex non-linear interaction seen between body mass and group size (*Janson & Goldsmith, 1995*). Given the links between blinking and communication discussed earlier, it is likely that social environment will influence blinking behaviour in species that are highly social (*Tada et al., 2013*), and there is much scope for comparative studies of this behaviour within and between species.

The reanalysis presented here did not show that blinking was related to diurnality (*Tada et al., 2013*), which may have been due to both the small number of nocturnal species considered ($n = 5$) and their high degree of nonindependence (where four of the five were from the Lorisoidea superfamily). It may be that a wider survey of nocturnal and cathemeral species may reveal that blinking is less common in darkness, and anecdotal results (*Stevens & Livermore, 1978*) suggest this result may hold true in non-primate species as well. If the major cost of not blinking is a reduction in eye health, then blinking behaviour is unlikely to differ between nocturnal and diurnal species.

Blinking is an understudied aspect of animal behaviour that could yield many new insights into the evolution of behaviour. Little is known about how blinking differs between species, and more detailed comparative studies comparing the behaviour both between and within species are needed, with larger sample sizes taken within species (as the current data were based on observations of between one and five individuals in captive conditions, which could conceivably mean that some of the results used were atypical for a species). This would need to be done using a suitably standardised technique (*Tada et al., 2013*) to allow comparison between species, as blinking behaviour is likely to be affected by environmental conditions (*Nakamori et al., 1997*; *Yorzinski & Argubright, 2019*; *Yorzinski, 2020a*). The reanalysis presented here highlights that care needs to be taken when considering data from related species, and that phylogenetically-controlled correlations may exist with ecological data from existing datasets. Once we understand how blinking varies within and between species, we can then use the behaviour as a tool to explore other aspects of an animal's world, such as its social biology (*Rowe, Robins & Rands, 2020*; *Tada et al., 2013*; *Matsumoto-Oda et al., 2018*) or its welfare (*Merkies et al., 2019*).

## ACKNOWLEDGEMENTS

I would like to thank two anonymous reviewers for their comments, which have greatly improved the manuscript.

### Funding
The author received no funding for this work.

### Competing Interests
The author declares that they have no competing interests.
## Author Contributions

- Sean A. Rands conceived and designed the experiments, performed the experiments, analysed the data, prepared figures and/or tables, authored or reviewed drafts of the paper, and approved the final draft.

## Data Availability

Raw data and code are available in the Supplemental Files.

## Supplemental Information

Supplemental information for this article can be found online at http://dx.doi.org/10.7717/peerj.10950#supplemental-information.

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
