# Peer review of "Phylogenetically-controlled correlates of primate blinking behaviour"

_PeerJ, doi:10.7717/peerj.10950_

## Round 0.1 · original submission · Major Revisions

Thank you for your submission to PeerJ. I have been fortunate to receive two reviews from experts who provide thoughtful and detailed feedback on your article.

While both reviewers note that the research question you pose is interesting and builds from previous work, both reviewers also asked for you to provide greater consideration over the factors that you include in your model, and I agree that this would be key. As noted by reviewer 2, you need to set out a clearly stated hypothesis in your introduction that paves the way for your model structure. I also urge you to more fully discuss the limitations of your approach, to consider alternative explanations for your results, and to propose future avenues for research in this area.

I look forward to receiving your revised article. When you prepare your revision, please provide an copy of your article showing tracked changes as well as point-by-point responses to each of the reviewers' comments and suggestions.

Reviewer 1 ·

Basic reporting

This paper reexamined factors influencing blinking behavior across primates. Using a previously published dataset, they used phylogenetically-controlled methods to examine the relationship between blinking behavior and locomotion, mass, group size, and trophic guild. Given that the previously published paper did not control for the phylogenetic relationships among species, this reanalysis is an important step in that direction. My main certain involves the selection of factors to consider. This paper uses some of the variables from the original dataset (body size, mass, activity) but not all (habitat type). It replaces the habitat type variable with locomotion data from a recently published database (Galan-Acedo) even though this database does include a habitat type variable. It would be helpful if more details could be provided on selection of variables.

Experimental design

no comment

Validity of the findings

no comment

Additional comments

Ln 27
“Performed” might be better than “conducted”

Ln 44
And peafowl (Yorzinski 2016)

Ln 53
What is meant by “other movement behaviors”?

Ln 72
“Account for” rather than “accommodate”

Ln 104
While Tada et al. termed the variable “Habitat Type,” they did group it into the same categories as you: terrestrial, arboreal, or a mix. How many species did you classify in the same way as Tada et. al? If you performed your phylogenetically-controlled analysis using their “Habitat Type” variable rather than your “Locomotion” variable, were the results qualitatively the same?

Galan-Acedo (2019) also present data on home range size and habitat type. Is there a reason you didn’t examine these variables? And, do their data for the other categories that you used from the Tada paper (i.e.., body mass and group size and activity) match?

Ln 106
Can you expand on this trophic guild more? What were the categories?

Ln 133
Why does locomotion appear twice in the model?

Ln 153
Galan-Acedo (2019) also present data on habitat type that seems like it could be used to directly examine habitat type to compare to Tada et al.

Ln 177
And chickens (Beauchamp 2017)

Reviewer 2 ·

Basic reporting

The manuscript is well written and well organized.

Experimental design

I have some major concerns that need to be addressed prior to publication:

1) The introduction leads to no hypotheses. The authors investigated three variables, namely the blink duration and rate, and tested the effect of several parameters. So, the study really needs clearly stated hypotheses, as some ecological conclusions are drawn in the discussion.

2) Because no hypothesis is stated, the authors don’t discuss crucial points highlighted by their results: namely the differences found between blink duration and blink rate.

3) The main aim of the study was to check the results of the original study of Tada et al. (2013) by taking into account phylogenetic similarity. However, the authors also changed the “habitat”/” locomotion mode” data. So, how to be sure that the differences found with the results of the original study are only explained by phylogeny? It is not clear what is the difference between the two datasets, and what it may involve.
[They also added a “trophic guild” parameter, but it was removed from the model because of non-significance].

4) In the introduction, the authors clearly show that blinking can be affected by the situation (i.e., risky, stressful…), the social and the environmental conditions. However, we don’t know in which condition the animals have been filmed. These are useful elements to mention, as it may cause a serious bias when recording blinking behavior from these videos. Moreover, it seems that the group size data have been harvested from the literature; they may not reflect the living conditions of the captive subjects at all…
Before comparing all these species with an ecological framework, it would have been relevant to check whether the investigated variables (i.e., blink duration and rate) are affected by the animal direct environment and activity (i.e., locomotion, rest). This would have helped to develop a standardized measurement of blinking behavior reducing all the potential biases.
That being said, I have really appreciated that the authors addressed these different issues in their discussion. And they provide good tracks to follow for further research!

5) I have been worried about the sample size / species: 1 to 6 individuals. It is quite small, especially if the blinking behavior is situation-dependent… and that people who recorded the data did not really control for it.

6) Finally, I would like to remind the authors of the broad readership of the PeerJ journal. Some important points/notions are not well defined, while some more details are required to allow the reader a good understanding. (see the General comments section)

Please, see the General Comments section for specific points to address.

Validity of the findings

As I previously said, my first concern is that the authors did not just take into account phylogenetic similarity, they also used another dataset regarding the locomotion mode… As it is not clear what is the difference between the original “habitat” dataset and the new “locomotion” dataset, and what it may involve, we can ask the following question: how to be sure that the differences found with the results of the original study are only explained by phylogeny?

Furthermore, regarding the effects of locomotion and group size: the conclusions have strong limitations due to the protocol used in the original study. A very small sample size per species, which is more concerning that if the blinking behavior is situation dependent, as it is suggested in the introduction, a great care should have been given to the experimental design. However, it is very vague in which condition animals have been filmed.
Also, group size data have been harvested from the literature. So, they may not reflect the living conditions of the captive subjects.

In addition, if the authors say that both blink rate and blink duration are related to locomotion mode, in fact, these results don’t go in the same direction.
“Blink duration was similar in solely terrestrial and solely arboreal species” L. 143
“Blink rate was lowest in arboreal, and highest in those that were solely terrestrial…” L.140
The authors need to provide precise hypothesizes in the introduction, and further discuss the differences they found between these two variables.

The variable having a powerful meaning, the trophic guild, had no effect at all…

For all these reasons, the results should be interpreted very cautiously… And I would not keep the present title. I would rather center on the need it highlights to consider phylogenetically-controlled correlations.

Additional comments

The main topic of the study is very interesting. The authors focus on the blinking behavior, which is a quite understudied aspect of animal behavior, while it may provide new insights into different aspect of the species’ ecology. They re-analyzed a previous study (Tada et al., 2013) by taking into account phylogenetic similarity. In fact, it changed the results, compared to the original study. They namely found the blinking rate and duration to be correlated with the locomotion mode.

So, this study deserves to be published, as it highlights a relevant point for further research: the real need to consider phylogenetically-controlled correlations when working with related species. However, I have some major concerns that need to be addressed prior to publication (please, see the Experimental Design section)

Below, the authors can find some points that are meant to help them improve this manuscript:

Abstract:

L9. The use of the term “terrestrial” may be confusing here. I suppose you use this word in opposition to marine species. But shortly after, you use it again but in opposition to arboreal (Cf. L.18).

L.13 « recorded and analyzed a comparative study »: the wording doesn’t make sense to me, could you rephrase it? Maybe: just “realized”. In fact, you analyzed their study.

L.14 “a large sample of captive primates”. As you talk later about phylogeny, you should specify that this sample includes many different primate species (it could be a large sample of individuals of a same species…).

L.22-24: What about the relation with body mass and group size? Could you add a few words about how these results could be interpreted?


Introduction:

L.37: What do you mean by “this blackout is not perceived”? Could you explain a little more ?

L.42 You should explain why it is relevant to investigate blinking behavior in some sleeping animals. It is not intuitive: as we close our eyes when sleeping, “blinking when sleeping” deserves some explanations. The PeerJ journal has a broad readership, which requires to provide more details.

L.53 “blinks are timed to occur alongside other movement behaviors”: What kind of movements? You should tell us more about it: are they movements related to social interactions, or locomotion? As in your study you test the effects of group size, and locomotion style, it would be interesting for the reader to know.

L.56 “which in turn will have been shaped by the selective processes that a species experienced during its evolution.” I am not sure of what you mean. Does “which” refer to “environmental and social conditions”? If this is the case, could you explain? Or please, rephrase this sentence.


L.58 I am not an English native speaker, but the wording sounds curious to me. « will be tied» … You should rather say: “…is likely tied to its ecology”, or “many evidences suggest that … is tied to its ecology”.

L.65 “suggested that diurnal species blinked more than nocturnal ones”: as you did not introduce this duality (i.e., diurnal vs. nocturnal) earlier, you should provide a suggestion about what could explain this difference.

L71-73: “Species that are closely related are likely to show similar behaviors and traits due to this recent history, and therefore any analyses that compare species need to accommodate this nonindependence of the datapoints”
Again, PeerJ is a generalist journal with a broad readership. I think you need to give the reader more keys to understand this notion. It is the crucial point of your paper, and why you have done this re-analyze. So, you definitely need to explain more and provide some examples.

Methods:

L.83 You use three different measures of blinking. You should clearly explain to the reader why it is relevant to take these three measurements of this same behavior: what is the difference between them, and what do you expect from that? Is there one standard measure used in previous studies, or always these three measures?
Maybe just say, for example: blinking/blackout can be reduced by 1) decreasing the rate, or 2) decreasing the duration, 3) decreasing both: so, you need to investigate these two parameters.
And you need to explain what do you expect from the third one.

L.89 “As well as providing these data describing blinking, Tada et al. (2013) also provided descriptive measures of the activity, habitat type, mean species group size and mean individual body mass, harvested from the literature.”
“Harvested from the literature”: do you mean that they did not take the group size and the body mass of the individuals they measured for blinking behavior? If this is the case, it should be clearly stated, as it is fundamental to interpret the results.
L.91 “In the current analysis, I use the original collected data for body mass (in kg) and group size”: are these data collected from the subjects, or the one harvested in the literature? It is not clear.

L.97-104: You should reread this paragraph:
“This classification of habitat [of Nada et al.] considered whether a species was primarily arboreal, semi-arboreal or terrestrial”
Vs.
“I instead collected data from Galan-Acedo et al. (2020) that described whether a species was primarily terrestrial, arboreal, or moved between both”
 With the elements that you provide, it is really hard for the reader to understand the difference between the two classifications… Please, rephrase it, or provide further explanations.

L.106 You should quickly define “Trophic Guild” for the broad readership of the journal. It is important because you integrated this variable in your model (Cf. L133). Also, did Tada et al. (2013) consider this variable?

L.133: “model “locomotion + mass + group size + trophic guild + locomotion”,
Why did you put « locomotion” twice?

L.134. “the least significant terms were then removed sequentially until the minimal adequate model was found.” Did you try the model selection approach using AIC?

Results:

Table1: In the caption, you should briefly provide the three modalities of the locomotion variable (i.e., terrestrial, arboreal, or moved between both, L.105), so the reader can understand what “both” means.

Discussion:

L.150 + L.166: The two variables blink rate and blink duration are not explained by the same parameters. There is a difference between both, could you provide a suggestion to explain it?
You found blink duration to be more sensitive to group size. It is interesting, as in the introduction, the result of the reference you provide about the effect of group size on blinking, is about blink rate (Rowe et al., 2020). What about olive baboons and chickens? (L.50-51) you just talk about a “reduction in blinking” but you don’t specify if it involves blink duration or blink rate …

L.149: Also, you say that “that both blink rate and blink duration are related to locomotion mode”, but, in fact, both results don’t go in the same direction.
“Blink duration was similar in solely terrestrial and solely arboreal species” L. 143
“Blink rate was lowest in arboreal, and highest in those that were solely terrestrial…” L.140
You definitely need to discuss this difference.

L.147 « the isolated blink ratio is not correlated with any of the descriptive measures considered”
Again, what did you expect from this variable, and why?

L. 150 “in species that spend at some of the time”

L. 170-172: “Standard theory about how vigilance levels are influenced by size of group that an individual is associated with, where individual vigilance will decrease as the number of co-vigilant group members increases”
The syntax of this sentence is not correct. Please rephrase it

L.184: How did you test for it (diurnal vs. nocturnal)?
In the model you provided “locomotion + mass + group size + trophic guild + locomotion” (L.133), I don’t see this parameter.

L.189 « If the major cost of not blinking is a reduction in eye maintenance »
I think the « not » is not correct, and leads to a contradiction.

---

## Round 0.2 · Minor Revisions

The two reviewers who reviewed your original submission have provided feedback on your revised article and both only provided minor suggested edits and clarifications. If you can address these few outstanding comments, it will be my pleasure to recommend your article for publication without further review.

Also, I wanted to note that Review 1 requested that you provide your dataset. I am sorry that they were not able to access it for review, but I wanted to confirm that I do see it with your submitted materials and was able to open and review it.

Reviewer 1 ·

Basic reporting

No comment

Experimental design

No comment

Validity of the findings

No comment

Additional comments

This resubmission provided additional details on the analysis. While it is still an interesting study, further interpretation of the results are still needed.

Specific Comments:

Line 17:
This implies that the only change in the analysis was in controlling for phylogeny.

Line 27:
Not all animals blink using an upper and lower lid (e.g., many birds blink side-to-side)

Line 40
This is only known to be the case in humans. There has not been tested in nonhumans.

Line 198
Can you create the violin plot for the blink duration vs. movement?

Line 204-207
This sentence is not completely logical. If the riskiest type of locomotion is arboreal then why would animals that are arboreal+terrestrial have the lowest blink durations? Your results seemingly provide conflicting information (less blinks in arboreal species but longer blinks). Does this mean that the total amount of time species spend blinking is similar, regardless of movement type? More discussion on this result and its possible interpretations are necessary.

Line 207
Contradict rather than “do not confirm”

Line 216
A recent study within a species provides support for the importance of blinking in locomotion: Yorzinski 2020, A songbird inhibits blinking behaviour in flight.

Line 222
What frame rate was used in the Tada study? Was it above the minimum blink duration?

Can you provide the raw data as a supplement?

Reviewer 2 ·

Basic reporting

The author has very kindly and carefully provided thorough responses to all my comments. The paper now is really fluid, very enjoyable to read. I would like to thank the author because it has been really easy to track the changes.
The additional explanations in the introduction much better lead to the analyzes that have been done in this study. Also, results are more clearly discussed, and limitations are well flagged in the discussion. I appreciate the author has chosen to focus on blink rate and blink duration (as the third blinking parameter was problematic). I am very glad the author has provided a new title, that really better announces and depicts the paper.

I have no outstanding concerns regarding the publication of this manuscript. This paper will be very helpful for future studies, on the specific blinking behaviour, but not only.

I am pleased that the authors have found my comments useful, and I have just a few more minor suggestions (Cf. Comments for the author).

Experimental design

No additional comments.
The author has provided thorough responses to my previous review.

Validity of the findings

No additional comments.
The author has provided thorough responses to my previous review.

Additional comments

I have just a few more minor suggestions:

(Line numbers refer to the annotated document).

Abstract
L.20: “Described by the locomotion style of a species, where species moving through arboreal environments blink least, ground-living species blink most, and species that use both environments show intermediate rates. The duration of a blink was also related to locomotion mode.”
—> Locomotion “style” / Locomotion “mode”. Also, in the Methods, you talk about Locomotion “environment”.
I think it would be better to use the same term. Using too different terms for the same variable can be confusing.

Intro
Maybe, it could be nice to add a final sentence at the end of the introduction:
Something like: The results will help to assess (1) the importance of controlling for phylogenetic similarity when considering the evolution of behavior, (2) whether blinking behavior could bring new insights into the ecology of species.

Methods
L. 157: isn’t --> is not
L. 158 : definition “of about” --> Definition of the

L.159 vs 161:
You are talking about one dataset but you refer to two papers Galán-Acedo et al 2020, 2019. That’s a little bit confusing, maybe you could integrate the two references differently.

---

## Round 0.3 · accepted · Accept

Thank you for responding to the reviewers' few outstanding comments. It is my pleasure to recommend your article for publication in PeerJ. Congratulations.